# Genome-Wide Identification of *HSF* Gene Family in Kiwifruit and the Function of *AeHSFA2b* in Salt Tolerance

**DOI:** 10.3390/ijms242115638

**Published:** 2023-10-27

**Authors:** Chengcheng Ling, Yunyan Liu, Zuchi Yang, Jiale Xu, Zhiyin Ouyang, Jun Yang, Songhu Wang

**Affiliations:** Anhui Province Key Laboratory of Horticultural Crop Quality Biology, School of Horticulture, Anhui Agriculture University, Hefei 230036, China

**Keywords:** kiwifruit, heat shock transcription factor, *AeHSFA2b*, transcriptional regulation, abiotic stress

## Abstract

Heat shock transcription factors (*HSFs*) play a crucial role in regulating plant growth and response to various abiotic stresses. In this study, we conducted a comprehensive analysis of the *AeHSF* gene family at genome-wide level in kiwifruit (*Actinidia eriantha*), focusing on their functions in the response to abiotic stresses. A total of 41 *AeHSF* genes were identified and categorized into three primary groups, namely, HSFA, HSFB, and HSFC. Further transcriptome analysis revealed that the expression of *AeHSFA2b/2c* and *AeHSFB1c/1d/2c/3b* was strongly induced by salt, which was confirmed by qRT-PCR assays. The overexpression of *AeHSFA2b* in Arabidopsis significantly improved the tolerance to salt stress by increasing *AtRS5*, *AtGolS1* and *AtGolS2* expression. Furthermore, yeast one-hybrid, dual-luciferase, and electrophoretic mobility shift assays demonstrated that *AeHSFA2b* could bind to the *AeRFS4* promoter directly. Therefore, we speculated that *AeHSFA2b* may activate *AeRFS4* expression by directly binding its promoter to enhance the kiwifruit’s tolerance to salt stress. These results will provide a new insight into the evolutionary and functional mechanisms of *AeHSF* genes in kiwifruit.

## 1. Introduction

The growth and development of plants are influenced by multiple abiotic and biotic stresses [1]. Especially, abiotic stress, including high temperature, high salinity, drought, cold, and other abiotic threats, is recognized as the leading cause of crop loss on a global scale, resulting in an approximate decline of 50% in crop productivity each year [2]. Presently, saline–alkali hazards have emerged as a widespread problem. Salinization of soil has the capacity to modify soil characteristics, decrease soil moisture content, and cause soil compaction, thereby impeding the growth and development of plants [3]. Plants are unable to alter their location to evade unfavorable stresses as opposed to animals, yet they have acquired particular adjustments over the course of evolution to confront these quickly changing stressors [4]. The tolerance of plants to stress is amplified by a multifaceted network of stress response mechanisms and a diverse range of adaptive mechanisms at the physiological, biochemical, and molecular levels, which enable them to adapt to changing environmental conditions [5]. Transcription factors (TFs), which are sequence-specific DNA-binding protein molecules, regulate the transcriptional expression of many downstream genes responsible for adverse stress [6,7,8]. For example, increasing reports have indicated that heat shock factor (HSF) proteins can activate *HSPs* gene expression by recognizing the HS elements (HSE; 5′-AGAAnnTTCT-3′) conserved in promoters of *HSPs* responses to heat defense [9]. In *Populus euphratica*, HSF binds to HSE cis-elements in the WRKY1 gene promoter, therefore regulating WRKY1 transcription to enhance transgenic Arabidopsis salt tolerance [10]. 

Heat shock transcription factors (HSFs) are a significant group of stress-responsive transcription factors (TFs) in eukaryotes [6]. They have been identified in various plant species subsequent to their initial discovery in yeast [11,12]. Research on diverse plants of the *HSF* gene family has demonstrated that the structure of the HSF protein is highly conserved, including conserved domains DNA binding domain (DBD) at the N-terminal, an adjacent bipartite oligomerization domain (HR-A/B or OD), nuclear localization signal (NLS), nuclear export signal (NES), C-terminal activator peptide protein (AHA), and repressor domain (RD) [13]. Three plant HSF classes: A, B, and C are identified based on number of amino acids between HR-A and HR-B regions [9]. Class A HSFs play a crucial role in activating transcription via AHA motifs, which consist of aromatic, hydrophobic, and acidic amino acid residues [14]. Conversely, these types of residues are not present in class B and C HSF. Moreover, the variable length of the linker between the DBD domain and HR-A/B region provides further evidence to support this categorization [13].

Plants synthesize a multitude of compatible molecules solutes, including mannitol, proline, and several soluble oligosaccharides (namely, Galactinol, trehalose, raffinose, and stachyose) to act as regulatory compounds for coping with various abiotic stress [15]. Under abiotic stress conditions, the raffinose family of oligosaccharides (RFOs), particularly raffinose, play a significant role in elevating the osmotic pressure within cells. Moreover, raffinose family oligosaccharides (RFOs) also serve multiple functions such as antioxidants, signaling molecules, facilitators of carbon transport and storage, and membrane trafficking agents to protect the plant cell from dehydration [16]. It is well known that Galactinol synthase (GolS; EC: 2.4.1.123) and Raffinose Synthase (RFS, EC 2.4.1.82), which are responsible for raffinose biosynthesis, play a key role in response to abiotic stresses [17,18]. Regarding rapeseed and tobacco, *BnGolS* and *NtGolS* genes have been well studied at a genome-wide level [19]. The mRNA expression level of *AtGolS1* and *AtGolS2* were expressed under the induction of drought and salinity; by contrast, *AtGolS3* was induced by cold stress but not by drought or salt stress [20,21]. Recently, Liu et al. reported that transgenic poplar of overexpression of *AtGolS2* and *PtrGolS3* enhanced salt stress tolerance compared with the control plants [22]. To date, there are a few reports regarding *RFS* genes in response to abiotic stresses. The mRNA expression level of *BvRS1* was rapidly induced by cold stress, whereas *BvRS2* was upregulated by salt stresses [23]. To conclude, the data indicate that the accumulation of galactinol and raffinose is associated with plant tolerance to abiotic stress.

The kiwifruit, belonging to *Actinidia*, contains a large genus containing more than 50 species that originated in China [24]. It is famous for its high vitamin C content and rich nutritional minerals that are beneficial for human health [25]. Significantly, abiotic stress can have a negative impact on both the yield and quality of kiwifruit. Some major varieties of kiwifruit exhibit a high level of sensitivity to salt stress, particularly exposed to NaCl [26]. Once the soil salinity concentration reaches 0.14%, it can result in salt damage to the plant and disrupt its regular growth and development. Furthermore, when the salt concentration rises to 0.54%, the kiwifruit yield will decrease sharply and the salinity may even lead to the death of the plant [27]. The problem of soil salinization in some kiwifruit production areas is becoming increasingly prominent, and salt stress has become the main obstacle to the sustainable development of the kiwifruit industry. 

In previous work, we comprehensively analyzed the evolution and expression patterns of the *RFS* and *GolS* gene family, and found that the transgenic Arabidopsis of overexpression of *AcRFS4* might enhance salt tolerance by modulation of raffinose content [28]. However, the mechanism underlying the regulation of *RFS* by transcription factors to enhance salt resistance remains unclear. In this study, gene structure, phylogenetic analyses, chromosomal location, gene duplication, and transcript profiles under abiotic stress of *AeHSF* gene family were performed. Moreover, *AeHSFA2b* has a strong correlation with *RFS*4 under salt stress conditions and was highly induced by salt stress. In addition, *AeHSFA2b* may bind to the HSE cis-elements site of the *AeRFS4* promoter by electrophoretic mobility shift assay (EMSA) and dual-luciferase reporter assay (LUC). Consistently, overexpression of *AeHSFA2b* increases *AtRS5*, *AtGolS1*, and *AtGolS2* expression level in transgenic Arabidopsis and increases salt tolerance. Taken together, *AeHSFA2b* plays a significant role in enhancing salt stress tolerance by positively regulating the raffinose synthesis via regulation of *RFS4*. The abovementioned results have significant theoretical and practical implications for improving our understanding of the molecular mechanism underlying salt tolerance in kiwifruit.

## 2. Results

### 2.1. Identification of the AeHSF Genes in Kiwifruit 

We identified 41 *AeHSF* genes from the kiwifruit database (http://kiwifruitgenome.org/ accessed on 2 January 2023) utilizing 20 *AtHSF* proteins from *Arabidopsis thaliana* as query sequences (Table 1). *AeHSFA1a-AeHSFC1b* were named according to the *AtHSF* gene names. Their typical characters of the *AeHSF* gene family amino acid sequences, including locus ID, linkage group distribution, the length of coding sequences, molecular weight (MW), and theoretical isoelectric point (pI), were detected (Table 1). A total of 11 proteins consisting of 400–700 amino acids (aa), 28 proteins containing 200–300 aa, and only 2 proteins with a length of less than 100 aa were observed. Thirteen *AeHSF* proteins were alkaline with PI values greater than 7.5, 23 proteins were acidic with PI values less than 5.1–7.0, and only 5 proteins were acidic with PI values less than 5.0.

### 2.2. Phylogenetic Structure and Motif of AeHSF Genes 

To further investigate the evolutionary relationships of *AeHSF*s, a total of 61 HSF protein sequences from Arabidopsis and kiwifruit were utilized to construct a phylogenetic tree. The 41 AeHSFs were categorized into three primary groups, namely, HSFA, HSFB, and HSFC (Figure 1). AeHSFA was the largest group with 26 members, which represented 63.4% of the total AeHSFs. The subsequent group was AeHSFB with 13 members, which accounted for 21.3% of the total AeHSFs. The smallest group was AeHSFC with only 2 members, which constituted only 4.8% of the total AeHSFs. In addition, the HSFA category is further divided into HSFA1-8 subgroups. The HSFB category is further divided into HSFB1-4 subgroups. The HSFC category is further divided into only one subgroup (Table 1). 

In order to understand the evolutionary relationship and the exon–intron organization of the *AeHSF* gene family, we conducted an analysis of multiple sequence alignment to 41 *AeHSF* protein sequences and the gene structure and conserved motif of *AeHSF* gene members. As shown in Figure 2A,B, we observed considerable variation in the number of introns present within the *AeHSF* genes, with a range from 1 to 5. The exon distribution analysis revealed that *AeHSF* genes within the same group exhibited a conserved gene structure and a similar number of exons, indicating highly conserved structures. In total, 10 motifs were predicted. Motif 1, Motif 2, and Motif 3 were identified as distinctive structural motifs of HSF. Motif 1 and Motif 2 were commonly found in all *AeHSF* genes, while in *AeHSFA3d*, Motif 3 existed alone (Figure 2C). Many *AeHSF* genes in the same groups had similar exon–intron structure and conserved motif, which was highly conservative in kiwifruit (Figure 2A–C).

### 2.3. Duplication and Chromosomal Location of the AeHSF Genes 

Gene duplication events are crucial in facilitating the emergence of novel functions and expanding the repertoire of genes. The duplication events of the HSF genes in the kiwifruit genome were analyzed. It was observed that no tandem duplication events occurred. However, a total of 28 pairs of segmental duplicates were identified (Figure 3A), suggesting that gene duplication serves as a fundamental mechanism for generating genetic diversity and facilitating the adaptation of organisms to changing environments. In *A. eriantha*, the positions of 41 *AeHSF* genes could be found on 20 chromosomes (Figure 3B). There was only one *AeHSF* gene on each of the chromosomes Chr4, Chr5, Chr6, Chr9, and Chr10. Five genes were located on chromosome 2, which contained the largest number of *AeHSF* genes in a chromosome (Figure 3B).

The 2000 bp sequences upstream promoter regions of the *AeHSF* genes were analyzed using PlantCARE (http://bioinformatics.psb.ugent.be/webtools/plantcare/html/ accessed on 4 June 2023). Some putative cis-elements responsive to stresses including W-box (defense and stress-responsive element), STRE (stress response element), DRE, CAAT--box (low-temperature responsive element), MBS (MYB binding site), and ABRE (ABA-responsive element) were found (Figure 4).

### 2.4. Expression of AeHSF Genes in Response to Abiotic Stresses and Hormone Treatments

Most of the *AeHSF* genes, with the exception of *AeHSFB4a-4c*, were significantly induced in response to abiotic stresses (Figure 5A). Noteworthily, 20 out of 41 *AeHSF* genes showed relatively higher expression levels after heat (HT) stress. Seventeen *AeHSF* genes were upregulated in response to drought (DT) stress. We also found that 18 *AeHSF* genes were expressed under NaCl stress. Among these genes, *AeHSFA2b*, *AeHSFA3C*, *AeHSFA5a*, *AeHSFA6a*, *AeHSFA1a-1d*, and *AeHSFA3a-3b* were strongly expressed, while the other genes were slightly expressed under NaCl stress. The gene *AeHSFA2a* was only expressed under waterlogging (WT) stress. As shown in Figure 5B, a significant induction was observed in about 61% of the *AeHSF* genes in response to ABA treatment. Especially, 10 *AeHSF* genes belonging to HSFB were strongly induced by ABA. Eight *AeHSF* genes were induced by GA. Only three *AeHSF* genes (*AeHSFA1b*, *A6a*, and *B1c*) had high expression in response to JA treatments. Only *AeHSFA1c*, *AeHSFA2c*, and *AeHSFA3a* were highly expressed after SA treatments.

### 2.5. Verification of Key AeHSF Genes Expression under NaCl Stress by qRT−PCR

To further validate the expression pattern of *key AeHSF* genes under salt stress, *AeHSFA2b/2c* and *AeHSFB1c/1d/2c/3b* genes were selected for qRT-PCR analysis. As shown in Figure 6, the expression levels of *AeHSFA2b/2c* and *AeHSFB1d/2c/3b* were significantly induced and peaked at 6 d under salt treatment. The expression levels of *AeHSFB1c* increased gradually during the whole salt-treated time. These results indicate that these genes were really induced by salt stress.

### 2.6. Correlation of All Transcription Factors with the Expressional Pattern of RFS4 under Salt Treatment and Transcriptional Activation of AeHSFA2b

The kiwifruit RNA-seqs of salt treatments were analyzed. The correlation results showed that *AeHSFA2b* transcription factors (TFs) had a strong correlation (R = approximately 0.95) with the expression patterns of *AeRFS4* under salt treatment (Figure 7A). The self-activation result of the Y1H Gold promoter showed that the growth of Y1H Gold containing the pABAi-*AeRFS4*-Pro recombinant plasmid was inhibited when exposed to a concentration of ≥200 ng mL^−1^ aureobasidin A (AbA) (Figure 7B). Only the positive control strain and transformed Y1H Gold harboring both pABAi-*AeRFS4*-Pro and pGADT7-*AeHSFA2b* could grow in a medium without leucine (-Leu) (200 ng mL^−1^ AbA), indicating protein–DNA interaction of *AeHSFA2b* and the *AeRFS4* promoter (Figure 7B).

### 2.7. Subcellular Localization of the AeHSFA2b Protein

Transient expression of the 35S::GFP plasmid demonstrated widespread GFP fluorescence across all cells, whereas the 35S:: AeHSFA2b:GFP plasmid exhibited distinct green fluorescence signals specifically localized in the nucleus (Figure 8). This finding suggests that *AeHSFA2b* likely encodes a nuclear-localized protein, which aligns with the functional attributes in its role of regulating gene transcription.

### 2.8. AeHSFA2b Regulates RFS4 Expression by Directly Activating and Binding to Its Promoter

In order to examine the regulatory role of AeHSFA2b on *AeRFS4* transcription, we conducted dual-luciferase reporter assays in tobacco protoplasts. The presence of the *RFS4*-pro LUC reporter and the 35S:AeHSFA2b-GFP effector resulted in a strong LUC fluorescence signal (Figure 9A). Conversely, the use of the 35S:GFP empty vector suppressed the fluorescence. Consistently, the LUC/REN ratio was significantly increased by the co-transformation of the *AeRFS4*-pro LUC reporter and the 35S:AeHSFA2b-GFP effector compared to the empty control (transformed with the reporter and the empty vector 35S:GFP) (Figure 9A). Moreover, we performed an electrophoretic mobility shift assay (EMSA) to further elucidate the specific and direct interaction between AeHSFA2b and the *AeRFS4* promoter. The EMSA result showed that the addition of increasing concentrations of cold probe resulted in a reduction in electrophoretic mobility shift from 0×–50×, suggesting that AeHSFA2b could physically bind to the HSE motif in the *AeRFS4* promoter to increase their expression (Figure 9B). Taken together, these findings indicate that AeHSFA2b can directly control the expression of *AeRFS4* by binding to the core HSE sequence (CTTGAAGCTTCAAG) located in the promoter regions of *RFS4*. 

### 2.9. The Effects of Overexpression of AeHSFA2b in Arabidopsis 

To identify functional characterization of *AeHSFA2b* under salt stress, three lines of transgenic Arabidopsis (designated as OE1, OE2, and OE3) were generated in this study. Under normal conditions (referred to as control), transgenic plants displayed robust growth without any noticeable differences. Compared to the wild-type, the transgenic plants of OE lines had significantly larger and heavier leaves after salt treatment (Figure 10A). The growth parameters (root length, plant fresh weight, and chlorophyll) of overexpressing the *AeHSFA2b* gene at seedlings were significantly higher than WT plants. The MDA contents of overexpressing the *AeHSFA2b* gene at seedlings were significantly lower than WT plants. These results indicate that *AeHSFA2b* overexpression indeed increased the tolerance to salt stress. 

Under salt stress, the expression of *AeHSFA2b*, *AtGolS1/2*, *AtRS5,* and salt-responsive marker genes expression of *AtHSP18.1*, *AtEGY3*, *AtRD29,* and *AtAPX2*, which is homologous to *AeRFS4*, in transgenic Arabidopsis lines exhibited a significant increase compared to the wild-type (Figure 11A,B), indicating that the *AeHSFA2b* transgenic Arabidopsis plants potentially improved salt tolerance possibly by promoting the accumulation of raffinose or other salt-responsive marker genes. 

## 3. Discussion

Heat shock factor (HSF) is an important family of transcription factors within the plant kingdom. In previous work, *HSF* genes have been identified as key players in safeguarding plants against abnormalities in growth patterns [29,30]. They play crucial roles in enabling plants to adapt to abiotic stress, and in maintaining the stability of cellular processes [1]. Over the past few years, the abundant availability of genomes has facilitated comprehensive research on the *HSF* gene family in many crop species, such a rice [31], tomato [32], wheat [33], pepper [34], potato [35], maize [36], etc. Nevertheless, there is currently no published research regarding the kiwifruit *HSF* gene family. In our study, we identified and classified 41 *AeHSF* genes in the kiwifruit genome. These genes were further categorized into three major classes (Table 1). We observed that the class A genes were the most abundant, totaling 24, followed by the class B members, which is consistent with findings in other plants [9]. Class A exhibited the highest diversity and abundance among all the identified clades, indicating significant evolutionary divergence and diversification (Figure 1). No HSF member belonging to the class A9 (At5g54070) was identified in kiwifruit. A similar result was found in both Chinese cabbage [37] and sesame [38]. In kiwifruit, a similar exon/intron structure was observed among most members of the *AeHSF* gene family within the same group, as evidenced by the phylogenetic relationships (Figure 2). These findings suggest that the *AeHSF* gene family has stayed relatively conservative during evolution.

In the former publication, the *HSF* gene was extensively studied for its capacity to specifically react to stress conditions caused by high temperatures in plants [11]. A total of 18 *PmHSF* gene members have been identified in *Prunus mume*, with 12 of them showing significant upregulation in response to high-temperature stress [39]. Under high-temperature and salt stress, the expression of genes belonging to the HSFA class, which constitutes one of the most active factors, exhibited a notable increase in *Sorbus pohuashanensis* [40]. The yield and quality of kiwifruit have been significantly impacted by abiotic stresses, such as salinity, drought, heat, cold, and waterlogging [26]. The *HSF* gene family orchestrates essential functions in diverse facets of plant growth and tolerance to abiotic stresses [9]. Nonetheless, only a limited number of studies have been conducted exploring the role of the *HSF* gene family in kiwifruit’s response to various abiotic stresses. In this study, analyses of the expression of *AeHSF* genes in kiwifruit were performed under different conditions: heat, low temperature, NaCl, waterlogging, UV, and drought, respectively. We founded that most of the *AeHSF* genes exhibited significant upregulation under heat stress, which is consistent with previous research [41]. Notably, AeHSFA2 and AeHSFA3 class gene members, with the exception of *AeHSFA2a*, were shown to be upregulated strongly under heat stress (Figure 5A). The role of *HSFA2* in the response to heat stress has been reported in kiwifruit and tomato [42,43]. *AeHSFB1b/1c/1d* and *AeHSFB3a/3b* were significantly induced after salt stress (Figure 5A). In Arabidopsis, the transcription of *AtHsfA2* was significantly induced under salt stress, and overexpression of *AtHsfA2* enhanced transgenic Arabidopsis’s tolerance to salt [44]. However, *OsHsfB2b* has been identified as a negative regulator of the stress response to tolerate drought and salt conditions in rice [45]. In addition, only *AeHSFA6a/6b/6c* was strongly induced after cold stress (Figure 5A). Huang et al. reported that the expression of *AtHSFA6b* is upregulated in response to cold stress and ABA hormone treatment in Arabidopsis [46]. Plant hormones are responsible for regulating the growth and development of plants, as well as response to stress, and play a vital role in a wide range of physiological and biochemical processes [47]. Here, the expression profiles of *HSF* genes under various hormone treatments were analyzed via transcriptome data. We found that the expression levels of approximately 61% of the *AeHSF* genes showed a significant increase in response to ABA treatment (Figure 5B). A similar result was reported by Zhang et al. [48]. The gene *HSFA6b* functions as a downstream regulator of this pathway, highlighting its significance in conferring heat stress resistance [46]. Interestingly, in Arabidopsis, it was observed that the expression of *HSFA6a* and *HSFA6b* was significantly decreased in response to both salt stress and abscisic acid (ABA) treatment [49]. Additionally, the expression of *HSFA1d/A6a*, and *HSFB1C* increased significantly under jasmonic acid (JA) hormone treatment (Figure 5B), which is similar to the previous research by Qi et al. [50]. In order to further investigate whether *AeHSF* genes are regulated by hormones, the cis-elements in the promoter regions of 41 *AeHSF* genes in kiwifruit were predicted. Twelve different types of putative cis-elements responsive to abiotic stresses and hormone were found in the 2000 bp promoters of *AeHSF* genes (Figure 4), which is consistent with that previously reported in tartary buckwheat and carnation [51,52]. It is worth mentioning that many *AeHSF* genes promote the contain abscisic acid responsive element (ABRE). Unsurprisingly, a similar result was found in many plants, such as *Brassica napus* [53], wild jujube [54], and *Arachis hypogaea* [55]. In short, these results indicate that the *AeHSF* gene family in kiwifruit plays a crucial role in response to abiotic stresses and hormones treatment. 

In recent years, multiple studies have consistently reported the function of *HSF* genes in many plants’ responses to abiotic stress. For example, the overexpression of *VpHSF1*, a member of the HSFB2 family, from Chinese Wild *Vitis pseudoreticulata* in tobacco revealed that *VpHSF1* exerts a dual regulatory role, acting as a positive regulator in acquired thermotolerance [56]. The overexpression of chickpea *CarHSFB2* in Arabidopsis has been shown to enhance the drought tolerance, which was achieved by promoting in the transcript levels of stress-responsive genes, specifically *RD22*, *RD26*, and *RD29A,* under drought stress conditions [57]. In wheat, *TaHSFB4-2B* transgenic Arabidopsis exhibited increased salt and drought tolerance [58]. The overexpression of the *TaHsfA6f* gene in Arabidopsis significantly enhanced tolerance to heat, drought, and salt stresses. Additionally, these transgenic plants exhibited heightened sensitivity to exogenous abscisic acid (ABA) [59]. However, a few reports have demonstrated how *HSF* genes regulate downstream stress-responsive genes to improve stress tolerance. In Arabidopsis, *AtHSFA2* can specifically regulate a subset of stress response genes, including *Hsp26.5*, *Hsp25.3*, *Hsp70b*, *APX2*, *RD29A*, *RD17*, *GolS1*, *IPS2*, *KSC1*, *ERD7*, and *ZAT10,* under heat stress [60]. Recently, the overexpression of *AcHsfA2-1* in kiwifruit was found to confer enhanced resistance to heat shock stress at 50 ℃ for a minimum of 2 h. It can bind to promoters of three specific *AcHsp20* genes (Acc16656, Acc25875, and Acc11386) to improve heat tolerance [42]. *AcHsfA2-1* in *A. chinensis* homologous gene is *AeHSFA2c* in *A. eriantha*, which was significantly induced after heat stress (Figure 5B). In the present study, we found that the *AeHSFA2b* transgenic Arabidopsis grew better than the wild-type under salt stress, and the expression level of gene related to raffinose synthesis was significantly increased (Figure 10A), indicating that the overexpressing transgenic plant reduced osmotic stress by increasing the content of raffinose, thus enhancing salt tolerance. Gu et al. suggested that the overexpression of *ZmHSFA2* in Arabidopsis resulted in an upregulation of these genes, *AtGOLS1*, *AtGOLS2*, and *AtRS5*. This led to an increase in raffinose content in leaves and improved tolerance to heat stress [61]. *RFS4* promoter contains a tandem inverted repeat of the HSE element. Apart from this, correlation analysis results indicated that *AeHSFA2b* and *RFS4* show a strong positive correlation under salt stress conditions (Figure 7A). Therefore, we speculated that the transcription factor *AeHSFA2b* may activate *RFS4* expression by directly binding its promoter to enhance the kiwifruit’s tolerance to salt stress. Yeast one-hybrid, dual-luciferase, and electrophoretic mobility shift assay results showed that *AeHSFA2b* is capable of binding to the HSE element on the promoter of *AeRFS4* in our study (Figure 7B and Figure 9). Similar results were found in Arabidopsis [62]. Lang et al. found that *BnHSFA4a* probably participates in dehydration tolerance by binding to the HSE elements of the *BhGolS4* promoter to regulate *BhGolS4* expression and the overexpression lines exhibited increased antioxidant abilities [63]. Overexpression of *AeHSFA2b* increases *AtRS5* (the homologous gene of *AeRFS4*), *AtGolS1,* and *AtGolS2* expression level in transgenic Arabidopsis and promotes salt tolerance (Figure 10B), suggesting that the regulating role of *AeHSFA2b* on raffinose might be conserved. In our previous study, overexpressing *AcRFS4* in Arabidopsis can enhance raffinose levels and increase the plant’s tolerance to salt stress. *AcNAC30* specifically interacted with the *AcRFS4* promoter, suggesting its involvement in regulating salt stress responses in kiwifruit plants [28]. Given the crucial roles of both *AcNAC30* and *AeHSFA2b* transcription factors in salt stress responses, it would certainly be valuable to discuss the potential crosstalk and interaction between two genes. This could involve exploring whether *AcNAC30* regulates the expression of *AeHSFA2b* genes or vice versa, and whether there are common *AcRFS4* or regulatory elements shared between two genes. Investigating this interplay could provide a deeper understanding of the regulatory networks involved in salt stress responses and contribute to the development of strategies for enhancing salt stress tolerance in kiwifruit and other crops.

## 4. Materials and Methods 

### 4.1. Identification and Phylogenetic Relationships Analysis of A. eriantha AeHSF Genes

The *Actinidia eriantha* (White) genome sequences and *AeHSF* protein family sequence were downloaded from the Kiwifruit Genome Database (http://kiwifruitgenome.org/, accessed on 10 March 2023) [64]. The profile hidden Markov model (PF00447) of HSF protein was obtained from Pfam32.0 database (http://pfam.xfam.org/, accessed on 12 March 2023), and all AeHSF protein sequences were screened in HMMER-3.3 (E-value < 1 × 10^−5^, other parameters set to defaults). All AtHSF amino acid sequences were downloaded from TAIR (https://www.arabidopsis.org/, accessed on 18 March 2023). The BLASTP program with a threshold of 1 × e^−5^ for the e-value was utilized to search the kiwifruit HSF protein using all AtHSF sequences amino acid as queries. The final members of the *AeHSF* gene family were identified through the combination of results obtained from both BLAST and HMMER results. We constructed the phylogenetic tree of related proteins using MEGA 11 software (bootstrap set at 1000, other parameters set to defaults) [65]. The phylogenetic tree was visualized and beautified using iTOL (https://itol.embl.de/, accessed on 25 March 2023) [66].

### 4.2. Gene Duplication and Chromosomal Locations

The *AeHSF* genes replication events were analyzed using multiple collinear scanning toolkits (MCScanX) with default parameters [67]. The physical location map of chromosomes was visualized by MapInspect 1.0. 

### 4.3. Expression Profiles Analysis of AeHSF Genes in Different Tissue, Hormone Treatments, and Abiotic Stress Treatments

The RNAseq raw sequence data of samples under the abiotic stress and hormone treatments were obtained from the NCBI website (Bioproject ID PRJNA1028382). RNA extraction, library preparation, and sequencing were conducted as previously described [68]. Trimmomatic was performed to filter the raw sequence data [69]. The clean data were adopted to map the reference *A. eriantha* variety ‘White’ genome using HISAT2 [70]. The genes were quantified with the featureCounts package in R. *A. chinensis* variety ‘Hongyang’ (HY) tissue culture seedlings were transferred to a soil mixture of perlite and sand (3:1, *v*/*v*). All seedlings were grown in a growth chamber at a temperature of 18 °C (night) and 24 °C (day), relative humidity of 60–80%, and a 14/10 h photoperiod (daytime, 06:00–20:00). The seedlings were irrigated with water once every two days. After two months, they were randomly divided into six groups for stresses treatments. For heat and cold stress treatment, the seedlings were transferred into two chambers with the temperature set at 48 °C and 4 °C, respectively. Treated seedlings were harvested at 6 h after treatment. For salt, drought, and waterlogging stresses, the seedlings were soaked in 0.6% NaCl for 6 days. The seedlings were flooded for 7 days, and the seedlings were dried for 14 days. Nontreated seedlings were used as the control (CK). TBtools software (version 2.012) was used to draw the heatmaps [71]. 

To further verify key *AeHSF* genes expression, the expressions of these genes were detected using qRT-PCR on the Biorad CFX96 real-time PCR system using the ChamQ SYBR qPCR Master Mix (Vazyme, Nanjing, Jiangsu, China). Total RNA isolation was carried out as reported previously. The relative gene expression level was determined using the 2^−∆∆Ct^ method. The primers for the nine *AeHSF* genes were designed using primer 3 software (http://frodo.wi.mit.edu/, accessed on 30 April 2023) (Appendix A). The bar chart was created with R software version 4.2.1 (geom_bar in ggplot2 library).

### 4.4. Correlation Analysis of All Transcription Factors with the Expressional Pattern of AeRFS4 under Salt Treatment and Transcriptional Activation Analysis of AeHSFA2b

A scatter plot graph of correlation all transcription factors with the expressional pattern of *AcRFS4* under salt stress was drawn with R software (geom_point in ggplot2 library). *AeHSFA2b* coding sequences were cloned into the pGBKT7 vector (Clontech, Madison, WI, USA) to create the pGBKT7-*AeHSFA2b* constructs, respectively (primers are listed in Appendix A). *AeHSFA2b* constructs and positive/negative control were transformed into yeast strain AH109. Transcription activation was measured according to a previous method [28].

### 4.5. Dual-Luciferase Reporter Assay and AeHSFA2b Protein Subcellular Localization Assay

The full-length coding sequence (CDS) of *AcHSFA2b* was inserted into the pGreenII 62-SK vector as an effector, while the promoter fragment of *RFS4* was cloned into the pGreenII0800-LUC vector to produce a reporter. *A. tumefaciens* strain EHA105 was utilized to transfer all the constructs vector (primers are listed in Appendix A). The effectors and reporters were transiently co-infiltrated into 4-week-old tobacco leaves. The suspension buffer for *A. tumefaciens* cells (10 mM MgCl_2_, 10 mM MES, 150 mM acetosyringone, with a pH of 5.6) was cultivated until it reached an OD600 of 0.6. After 2–3 d of injection at 23 °C, tobacco leaves were sprayed with water for the control, and the experimental leaves were treated with 1000 mg/kg ethephon. The promoter activities expressed as a ratio (LUC/REN) were using a dual-luciferase kit (YEASEN, Shanghai, China), and determined by a microporous plate light detector (Berthold Centro LB960). Fluorescence in tobacco leaves was observed and photographed using a fully automated chemiluminescence imager (Tanon 5200, Shanghai, China).

The CDS of *AeHSF2b* without the stop codon was constructed into the vector pCAMBI A1305-35S-GFP vector using double-enzyme digestion (cut with Xbal and BamHI). Then, the plasmid (35S::AeHSF2b-GFP) was transferred into Agrobacterium strain EHA105. This strain was injected into 1-month-old tobacco leaves. Finally, the GFP fluorescence was detected by confocal laser scanning microscopy. Primers used to construct the 35S::AeHSF2b-GFP vector are provided in Appendix A.

### 4.6. EMSA Protein Expression and Electrophoretic Mobility Shift Assay (EMSA)

The ORF of *AeHSFA2b* was cloned into the pGEX-4T-1 vector containing GST (primers are listed in Appendix A), followed by transformation into competent cells of *E. coli* BL21 (DE3) (TransGen, Beijing, China) to generate recombinant AeHSFA2b-GST protein. The recombinant GST- AeHSFA2b *E. coli* cells were cultivated until they reached an OD600 of 0.6 at 37 °C, and then IPTG was added to 0.5 mM for a final concentration. Recombinant protein expression was induced for a period of 10–12 h at 28 °C and was purified using GST fusion protein purified magnetic beads kit (Beaver, Suzhou, Jiangsu, China) according to the manufacturer’s instructions. The probe sequence (Appendix A) containing the *AeHSFA2b* binding sequences of *AeRFS4* was predicted by PlantCare (http://bioinformatics.psb.ugent.be/webtools/plantcare/html/, accessed on 15 April 2023). The probes were synthesized Tsingke Biotechnology (Nanjing, Jiangsu, China) and labeled with EMSA Probe Biotin Labeling Kit (Beyotime Biotechnology, Nantong, Jiangsu, China). The Chemiluminescent EMSA Kit (Beyotime Biotechnology) was utilized to perform a DNA gel mobility shift assay following the manufacturer’s instructions.

### 4.7. AeHSFA2b-Overexpressing Arabidopsis Generation and Salt Stress Tolerance Analysis

The ORF of the *AeHSFA2b* was cloned into pCAMBIA1305 by the primers (Appendix A). The flower dip method was employed to transform the confirmed constructs (AeHSFA2b-1305) into Arabidopsis ecotype Columbia plants (Col-0) via Agrobacterium-mediated transformation. The transgenic Arabidopsis lines (OE1, OE2, and OE3) were distinguished on 1/2 MS medium supplemented with 25 mg·L^−1^ Hygromycin B (HYG). Homozygous T3 seedlings of *A. thaliana* from transgenic lines and wild-type (WT) seedlings aged five weeks were individually grown on 1/2 MS medium containing 150 mmol·L^−1^ NaCl for 7 days. After 7 days, the root lengths of WT and transgenic Arabidopsis were measured by using ImageJ 1.8.0. The leaves of WT and transgenic Arabidopsis under salt stress treatments were collected. Physiological and biochemical indicators were measured using corresponding assay kits following the manufacturer’s protocols. In detail, the Chlorophyll Assay Kit (Solarbio, Cat#BC0990, Beijing, China) and Malondialdehyde (MDA) Assay Kit (Solarbio, Cat#BC0020) were used for subsequent spectrophotometric. *AeHSFA2b, AtRS5*, *AtGolS1*, and *AtGolS2* genes were selected for qRT-PCR analysis (primers are listed in Appendix A). In addition, the salt-responsive marker gene expression levels in the *AeHSFA2b* transgenic lines, such as *AtHSP18.1*, *AtDREB2A*, *AtRD29,* and *AtAPX2* genes, were selected for qRT-PCR analysis (primers are listed in Appendix A).

## 5. Conclusions 

Overall, we conducted a comprehensive analysis of *AeHSF* family members in kiwifruit at the genome-wide level and successfully classified these members into three distinct groups based on the conserved domains. The transcription profile of the *AeHSFA2b* gene provided valuable insights into its potential role in enhancing salt tolerance. The ectopic expression technique was employed to investigate the impact of *AeHSFA2b* on salt stress tolerance in Arabidopsis. Moreover, yeast one-hybrid, dual-luciferase, and electrophoretic mobility shift assay results indicated that AeHSFA2b is capable of binding to the HSE element on the promoter of *AeRFS4*. Together, our research provides a foundational framework for further exploration of the functional significance of *HSF* genes in enhancing kiwifruit’s tolerance to abiotic stresses.

## Figures and Tables

**Figure 1 ijms-24-15638-f001:**
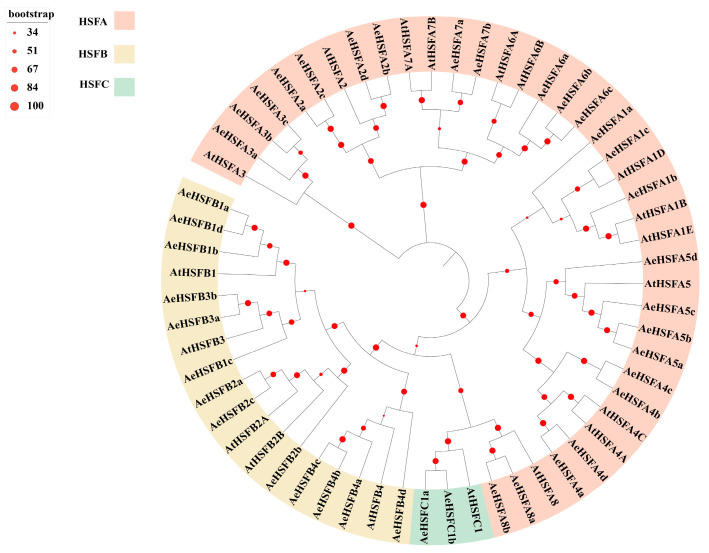
Phylogenetic analysis of *A. eriantha* (AeHSFs) and *A. thaliana* (AtHSFs) proteins. The tree was constructed using MEGA 6.0 by the neighbor-joining method with 1000 bootstrap replicates. The number on the branch denotes the reliability of the node based on 1000 bootstrap verification; Different colors of the inner ring indicate different classification results for HSFs.

**Figure 2 ijms-24-15638-f002:**
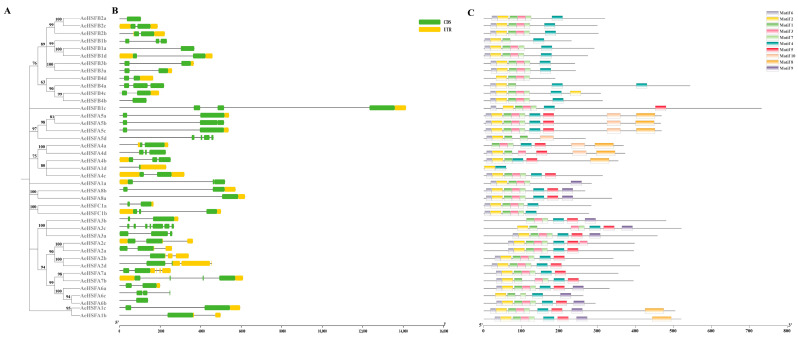
Analysis of the gene structure and conserved motif of *AeHSF* gene members. (**A**) Neighbor-joining phylogenetic tree of *AeHSF* members. (**B**) The gene structure of the *AeHSF* genes. Green boxes represent exons (CDS), black lines represent introns, and yellow boxes represent 5′ and 3′ untranslated regions. (**C**) The conserved motif of *AeHSF* genes. In total, 10 motifs were identified. The protein length is shown on the horizontal scale. The vertical scale indicates the sequence identity of each conserved motif.

**Figure 3 ijms-24-15638-f003:**
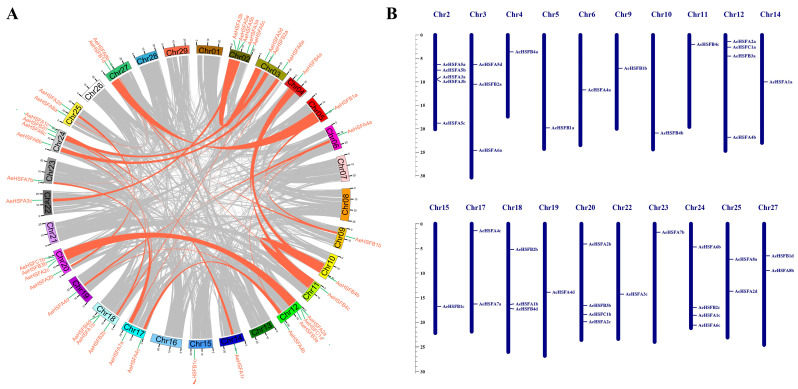
Gene duplication analysis and chromosomal locations of *AeHSF* genes. (**A**) Gene duplication analysis of *AeHSF* genes. Pink lines connect the syntenic regions between kiwifruit *AeHSF* genes. (**B**) Chromosomal locations of *AeHSF* genes in *A. eriantha*. The chromosome number is marked on each chromosome.

**Figure 4 ijms-24-15638-f004:**
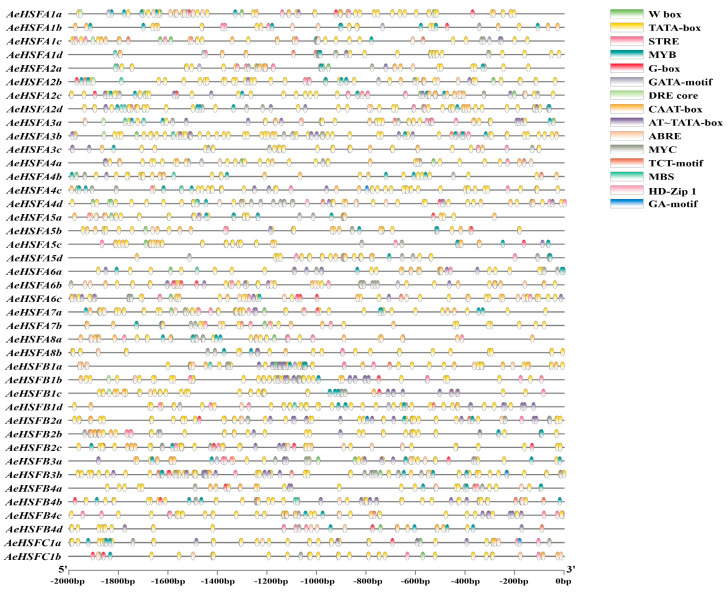
Predicted cis–elements in the promoter upstream 2000 bp sequences of *AeHSF* genes, which are presented as colored rectangles.

**Figure 5 ijms-24-15638-f005:**
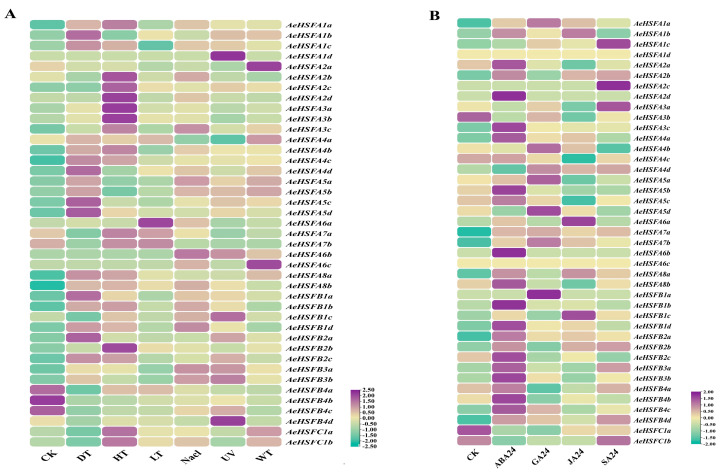
The expression analysis of *AeHSF* genes under abiotic stresses and hormone treatments. (**A**) Expression profiles of *AeHSF* genes under abiotic stresses. (**B**) Expression profiles of *AeHSF* genes under abiotic stresses under hormone treatments. The color bar to the right of the figures represents the log2 FPKM (fragments per kilobase of exon per million reads mapped) value.

**Figure 6 ijms-24-15638-f006:**
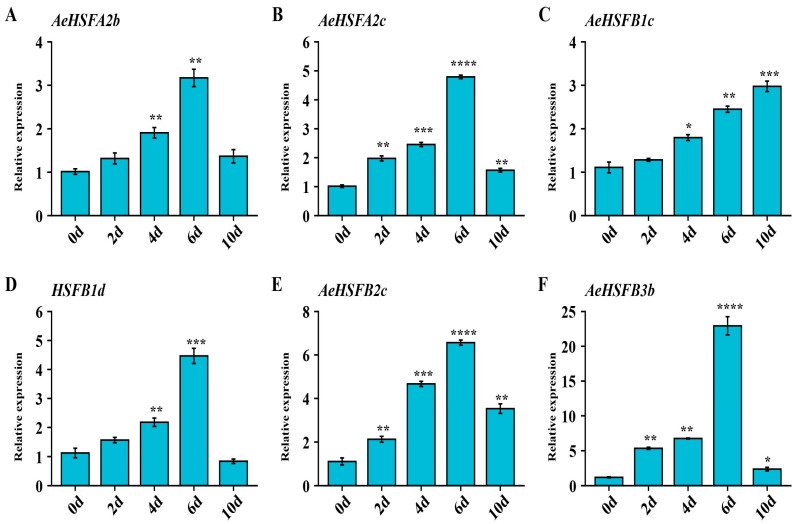
Expression levels of key *AeHSF* genes under salt stress. Three biological and technical replicates calculated the error bars. Asterisks indicate the corresponding gene significantly up- or downregulated under the different treatments using *t*-test (* *p* < 0.05, ** *p* < 0.01, *** *p* < 0.001, **** *p* < 0.0001).

**Figure 7 ijms-24-15638-f007:**
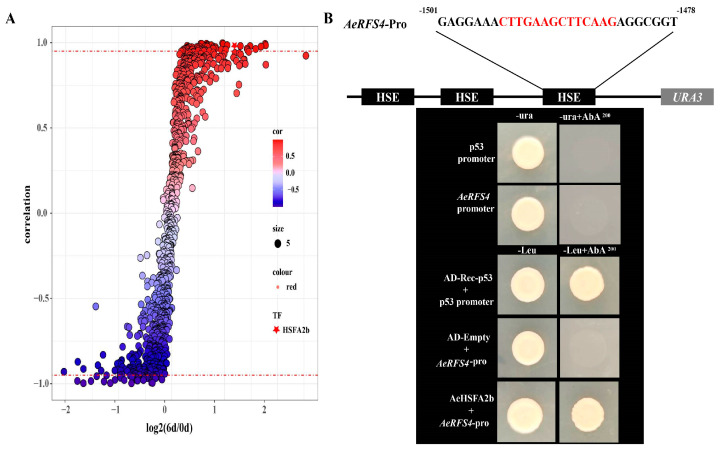
*AeHSFA2b* transcription factor in the modulation of AeRFS4. (**A**) Correlation analysis of all transcription factors with the expressional pattern of *AeRFS4* based on the RNA−seq results. The x axis indicates log2 (FPKM ratio of salt 6 d: CK 0 d). The y axis indicates the values of correlation between *RFS4* and each transcription factor. Pentangles show the *AeHSFA2b* that were highly correlated (R ≥ 0.95, red dashed lines) with the expression patterns of *RFS4.* (**B**) Interaction of *AeHSFA2b* with the promoter of RFS4 in the Y1H assay.

**Figure 8 ijms-24-15638-f008:**
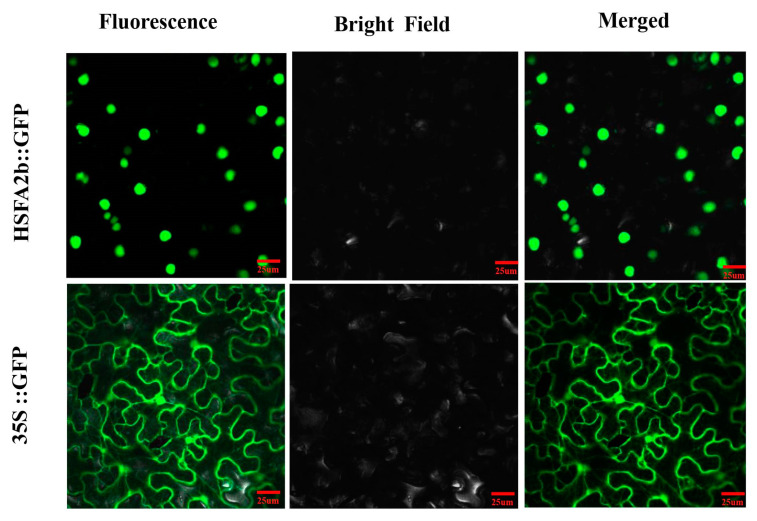
Subcellular localization of *AeHSFA2b* and empty vector in *Nicotiana benthamiana* leaves after 2 days of infiltration. The green fluorescence of GFP is indicated. White arrows indicate the GFP signal from the nucleus. Scale bars = 25 μm.

**Figure 9 ijms-24-15638-f009:**
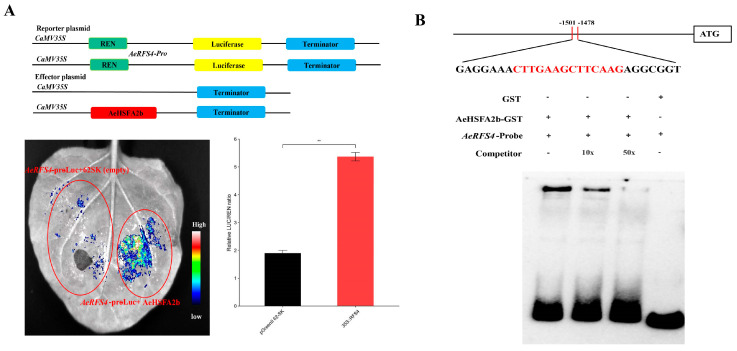
The direct targeting and transcriptional modulation of the *RFS4* promoter by *AeHSFA2b*. (**A**) A dual−luciferase (LUC) assay in *Nicotiana benthamiana* leaves was performed to explore the transcriptional activation function of *RFS4* on *AeHSFA2b* promoters. The control group for the experiment involved the use of an empty vector. Representative images were captured to visually represent the outcomes, and the activity of LUC/Renilla luciferase (REN) was quantified to validate the transcriptional activation of *RFS* by *AeHSFA2b*. The error bars in the graph indicate the standard deviation (SD). Statistical analysis was conducted using the Student’s *t*-test. Significance at *p* < 0.01 is denoted by **. (**B**) The AeHSFA2b protein exhibited binding affinity towards specific regions of the *AeRFS4* promoter by using an electrophoretic mobility shift assay (EMSA).

**Figure 10 ijms-24-15638-f010:**
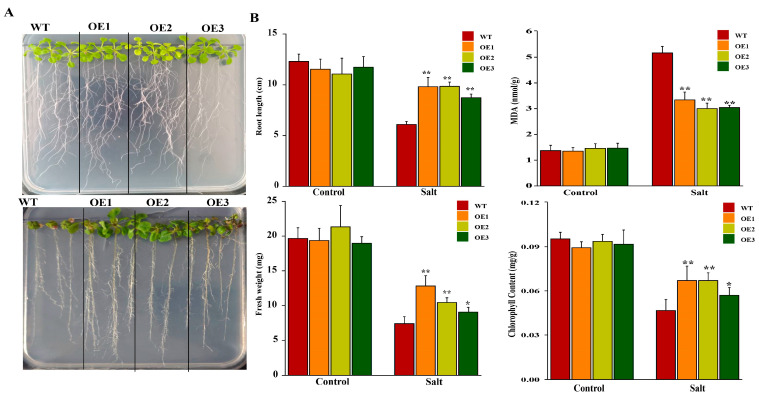
Phenotypic and genes expression T3 *AeHSFA2b*-transgenic *Arabidopsis* seedlings under different NaCl concentrations. (**A**) Observation of phenotype, the root length, and fresh weight under different NaCl concentrations NaCl. (**B**) Genes expression of *AeHSFA2b, AtGolS1/2,* and *AtRS5* under different NaCl concentrations. Three biological and technical replicates calculated the error bars. Asterisks indicate the corresponding gene significantly up- or downregulated under the different treatments using *t*-test (* *p* < 0.05, ** *p* < 0.01). WT, wild-type *Arabidopsis*. OE1, OE2, and OE3, three different lines of transgenic *Arabidopsis*.

**Figure 11 ijms-24-15638-f011:**
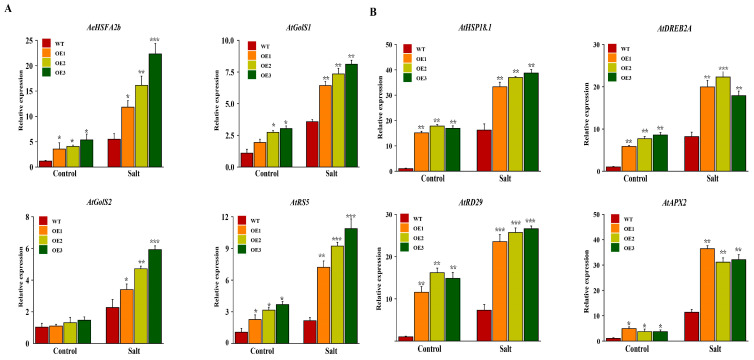
Genes expression T3 *AeHSFA2b*-transgenic *Arabidopsis* seedlings under different NaCl concentrations. (**A**) Genes expression of *AeHSFA2b*, *AtGolS1/2,* and *AtRS5* under different NaCl concentrations. (**B**) Salt-responsive marker genes expression of *AtHSP18.1*, *AtEGY3*, *AtRD29,* and *AtAPX2* under different NaCl concentrations. Three biological and technical replicates calculated the error bars. Asterisks indicate the corresponding gene significantly up- or downregulated under the different treatments using *t*-test (* *p* < 0.05, ** *p* < 0.01, *** *p* < 0.001). WT, wild-type *Arabidopsis*. OE1, OE2, and OE3, three different lines of transgenic *Arabidopsis*.

**Table 1 ijms-24-15638-t001:** Detailed information on 41 *AeHSF* genes in kiwifruit.

Gene Name	Genome ID	Chr ID	Start (bp)	End (bp)	Protein Length (aa)	Molecular Weight (kDa)	PI
DTZ79_14g07720	*AeHSFA1a*	Chr14	10,001,679	10,006,882	283	31,275.38	5.47
DTZ79_18g08160	*AeHSFA1b*	Chr18	16,404,715	16,409,700	503	55,972.83	6.19
DTZ79_24g12140	*AeHSFA1c*	Chr24	18,591,496	18,597,442	512	56,190.77	7.76
DTZ79_12g13240	*AeHSFA1d*	Chr12	23,908,734	23,911,031	59	6772.63	7.83
DTZ79_12g01030	*AeHSFA2a*	Chr12	1,389,789	1,392,367	394	45,138.73	4.53
DTZ79_20g03210	*AeHSFA2b*	Chr20	4,073,776	4,077,184	340	39,266.25	6.43
DTZ79_20g13080	*AeHSFA2c*	Chr20	19,899,248	19,902,860	396	45,173.66	10.25
DTZ79_25g04820	*AeHSFA2d*	Chr25	13,660,194	13,664,745	410	47,170.3	6.05
DTZ79_02g08920	*AeHSFA3a*	Chr02	9,444,887	9,447,485	456	51,324.32	5.1
DTZ79_02g08960	*AeHSFA3b*	Chr02	9,502,675	9,505,569	479	53,705.08	8.16
DTZ79_22g06060	*AeHSFA3c*	Chr22	14,257,374	14,260,052	519	58,377.59	6.79
DTZ79_06g06820	*AeHSFA4a*	Chr06	11,684,714	11,687,108	367	41,842.63	7.87
DTZ79_12g12240	*AeHSFA4b*	Chr12	21,843,255	21,845,772	353	39,637.21	5.12
DTZ79_17g00990	*AeHSFA4c*	Chr17	1,375,483	1,378,676	312	36,435.81	5.95
DTZ79_19g05240	*AeHSFA4d*	Chr19	13,922,263	13,924,540	371	42,169.33	8.3
DTZ79_02g06560	*AeHSFA5a*	Chr02	6,267,378	6,272,776	467	52,808.94	5.6
DTZ79_02g07500	*AeHSFA5b*	Chr02	7,451,281	7,456,434	465	52,607.75	9.15
DTZ79_02g14180	*AeHSFA5c*	Chr02	18,802,710	18,808,087	467	52,763.84	8.93
DTZ79_03g06230	*AeHSFA5d*	Chr03	6,337,642	6,342,273	267	29,710.02	4.62
DTZ79_03g18750	*AeHSFA6a*	Chr03	24,588,106	24,590,103	330	38,174.39	4.91
DTZ79_24g03470	*AeHSFA6b*	Chr24	4,741,552	4,742,952	293	34,385.11	6.37
DTZ79_24g13410	*AeHSFA6c*	Chr24	20,568,827	20,571,324	261	30,050.22	7.53
DTZ79_17g08930	*AeHSFA7a*	Chr17	16,311,605	16,314,121	353	40,323.58	6.76
DTZ79_23g01840	*AeHSFA7b*	Chr23	1,719,685	1,725,774	393	45,040.76	5.97
DTZ79_25g01580	*AeHSFA8a*	Chr25	7,169,500	7,175,687	336	38,474.7	5.11
DTZ79_27g07960	*AeHSFA8b*	Chr27	9,529,778	9,535,497	266	30,879.35	5.970
DTZ79_05g10500	*AeHSFB1a*	Chr05	19,792,047	19,795,733	290	32,502.43	5.84
DTZ79_09g05760	*AeHSFB1b*	Chr09	7,085,426	7,087,756	230	25,915.03	6.72
DTZ79_15g12130	*AeHSFB1c*	Chr15	16,819,805	16,833,941	730	83,846.59	8.83
DTZ79_27g06080	*AeHSFB1d*	Chr27	6,538,958	6,543,540	273	30,221.65	5.16
DTZ79_03g10230	*AeHSFB2a*	Chr03	10,487,781	10,488,831	318	35,100.92	4.54
DTZ79_18g03510	*AeHSFB2b*	Chr18	5,178,304	5,180,532	301	33,379.45	6.3
DTZ79_24g11010	*AeHSFB2c*	Chr24	17,046,072	17,047,945	298	33,423.12	8.23
DTZ79_12g03600	*AeHSFB3a*	Chr12	4,511,163	4,513,746	241	27,528.89	7.55
DTZ79_20g10710	*AeHSFB3b*	Chr20	16,559,515	16,563,171	239	27,835.34	6.02
DTZ79_04g03520	*AeHSFB4a*	Chr04	3,612,012	3,614,202	542	60,737.75	5.93
DTZ79_10g10590	*AeHSFB4b*	Chr10	20,942,779	20,944,104	312	35,352.01	5.65
DTZ79_11g01990	*AeHSFB4c*	Chr11	2,045,916	2,047,860	307	34,740.27	4.99
DTZ79_18g08770	*AeHSFB4d*	Chr18	17,220,840	17,222,490	187	21,116.75	6.72
DTZ79_12g02240	*AeHSFC1a*	Chr12	2,633,942	2,635,609	282	32,163.46	7.69
DTZ79_20g11770	*AeHSFC1b*	Chr20	18,439,575	18,444,574	276	31,061.17	5.88

## Data Availability

The datasets generated are available from the National Center for Biotechnology Information repository (https://dataview.ncbi.nlm.nih.gov/object/PRJNA1028382).

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
