# Peer review of "Genome-Wide Identification of *HSF* Gene Family in Kiwifruit and the Function of *AeHSFA2b* in Salt Tolerance"

_ijms, 2023, doi:10.3390/ijms242115638_

Round 1
Reviewer 1 Report
Comments and Suggestions for Authors
The authors comprehensively analyzed AeHSF family members in kiwifruit at the genome-wide level and classified these members into 3 distinct groups. The transcription profile of the AeHSFA2b gene has provided valuable insights into its potential role in enhancing salt tolerance. Generally, the manuscript is well-written, the methodology used appears to be correct, and the results obtained are presented clearly. In my opinion, this research can provide a foundational framework for further exploration of the functional significance of HSF genes in enhancing kiwifruit's tolerance to abiotic stresses.
Some minor revision is needed here as follow:
1) Sentences cannot begin with an abbreviation. More sentences must be rewritten from this point of view ( see the sentences from the following rows: R15;116, R177, R239, R 255, R325, R529);
2) The common name of plants must be writted with a small letter (for example: instead of Rapeseed must be written rapeseed, etc.)
3) In the manuscript the authors do not refer to figure 2A.
he authors comprehensively analyzed AeHSF family members in kiwifruit at the genome-wide level and classified these members into 3 distinct groups. The transcription profile of the AeHSFA2b gene has provided valuable insights into its potential role in enhancing salt tolerance. Generally, the manuscript is well-written, the methodology used appears to be correct, and the results obtained are presented clearly. In my opinion, this research can provide a foundational framework for further exploration of the functional significance of HSF genes in enhancing kiwifruit's tolerance to abiotic stresses.
Some minor revision is needed here as follow:
1) Sentences cannot begin with an abbreviation. More sentences must be rewritten from this point of view ( see the sentences from the following rows: R15;116, R177, R239, R 255, R325, R529);
2) The common name of plants must be writted with a small letter (for example: instead of Rapeseed must be written rapeseed, etc.)
3) In the manuscript the authors do not refer to figure 2A.
Author Response
Reviewer, IJMS
October 19, 2022
Dear Reviewer 1
Thank you very much for your valuable comments and suggestions. We have revised manuscript according to the suggestions. We also give point-to-point answers to all the questions.
1、Sentences cannot begin with an abbreviation. More sentences must be rewritten from this point of view (see the sentences from the following rows: R15;116, R177, R239, R 255, R325, R529);
Respond: Thank you for the comments. The sentences mentioned in the rows R15, R116, R177, R239, R255, R325, and R529 have been rewritten accordingly.
2、The common name of plants must be writted with a small letter (for example: instead of Rapeseed must be written rapeseed, etc.)
Respond: Thank you for bringing this to our attention. We have corrected this error and ensured that all common names are written with lowercase letters.
3、In the manuscript the authors do not refer to figure 2A.
Respond: Thank you for pointing out this oversight. We added the reference to Figure 2A in the right place of our revised manuscript.
Best regards,
Jun Yang

Reviewer 2 Report
Comments and Suggestions for Authors
In this manuscript, Ling et al. identified the Heat shock transcription factors in kiwifruit and characterized one of them, AeHSF2b, in salt stress response. The authors identified 41 AeHSF genes and examined their expression patterns based on RNA-seq and qRT-PCR experiments. To characterize the biological function of salt-responsive AeHSF2b, they examined the potential target of AeHSF3b and phenotypes of overexpression transgenic lines in the heterologous system of Arabidopsis. The following are my major comments on this manuscript.
1. RNA-seq data: The main findings of this manuscript are based on the expression pattern analysis of RNA-seq data. However, the authors did not share the raw data and the methods for data analysis. I strongly recommend depositing the raw data to the public database such as NCBI GEO and provide the detailed methods how the data were analyzed.
2. The detailed methods: The authors did not provide detailed information on the methods. For instance, the salt concentration and duration of stress treatment are missing in the manuscript.
3. Phenotype of the salt stress response: The authors only show the root length of overexpression lines to examine the salt tolerance. More supporting data, such as fresh weight, photosynthesis efficiency, and chlorophyll contents are required to convince the salt-tolerant phenotype. In addition, it is recommended to examine the salt-responsive marker gene expression levels in the transgenic lines.
4. Introduction and Discussion: It will be better to explain why the study on the salt stress response in kiwifruit is important in the introduction. In the previous paper, the authors reported that AcNAC30 could bond with the AcRFS4 promoter. It will be better to discuss the mechanistic insight on the NAC and HSF in the salt stress response.
Author Response
Reviewer, IJMS
October 19, 2022
Dear Reviewer 2
Thank you very much for your valuable comments and suggestions. We have revised manuscript according to the suggestions. We also give point-to-point answers to all the questions.
- RNA-seq data: The main findings of this manuscript are based on the expression pattern analysis of RNA-seq data. However, the authors did not share the raw data and the methods for data analysis. I strongly recommend depositing the raw data to the public database such as NCBI GEO and provide the detailed methods how the data were analyzed.
Respond: Thank you for your suggestion. The raw RNA-seq data generated from this study have been deposited in the NCBI (PRJNA1028382,Reviewer link: https://dataview.ncbi.nlm.nih.gov/object/PRJNA1028382?reviewer=u16cm3ham2n0gd5um11smtpm9d), which will be released soon. The RNA-seq data analysis pipeline employed in this study involved several steps, including quality control, read alignment, quantification of gene expression levels, and statistical analysis. We elaborate on these steps in the revised manuscript, providing a clear and concise description of the computational methods in Line444-451.
- The detailed methods: The authors did not provide detailed information on the methods. For instance, the salt concentration and duration of stress treatment are missing in the manuscript.
Respond: Thank you for the comments. we provided comprehensive descriptions of the methods in the revised manuscript; we will outline the specific salt concentration used and the duration of the stress treatment for each experimental condition from Line521-522. we have indeed provided specific details of sample processing and methods for transcriptome analysis. These details include the steps involved in RNA extraction, library preparation, and sequencing, along with the specific platforms and software used for data analysis from Line429-434. We have also supplemented specific methods for subcellular localization from Line493-498 and the determination of physiological indicators from Line522-531.
- Phenotype of the salt stress response: The authors only show the root length of overexpression lines to examine the salt tolerance. More supporting data, such as fresh weight, photosynthesis efficiency, and chlorophyll contents are required to convince the salt-tolerant phenotype. In addition, it is recommended to examine the salt-responsive marker gene expression levels in the transgenic lines.
Respond: We appreciate your valuable feedback and suggestions regarding the phenotype of the salt stress response in our study. We agree that additional supporting data would strengthen our conclusions and provide a more comprehensive assessment of the salt-tolerant phenotype. We have included additional parameters such as root length, fresh weight, MDA, and chlorophyll contents in our analysis to further evaluate the salt tolerance of the overexpression lines in the figure 10. These measurements provided more details of the plant's physiological response to salt stress. Furthermore, we have also examined the expression levels of salt-responsive marker genes (AtHSP18.1, AtRD29, AtDREB2A and AtAPX2) in the transgenic lines in the figure 11. The selection of salt-responsive marker was based on the literature (The Heat Stress Factor HSFA6b Connects ABA Signaling and ABA-Mediated Heat Responses), which also showed the role of HSFA6 in the salt stress. Due to the limited resources in our laboratory, we lack the sophisticated equipment to accurately measure the photosynthesis of the small leaves of our young Arabidopsis seedlings in the petri dish.
- Introduction and Discussion: It will be better to explain why the study on the salt stress response in kiwifruit is important in the introduction. In the previous paper, the authors reported that AcNAC30 could bond with the AcRFS4 promoter. It will be better to discuss the mechanistic insight on the NAC and HSF in the salt stress response.
Respond: Thank you for your valuable comments. We appreciate your suggestion to provide a better explanation of the importance of studying the salt stress response in kiwifruit in the introduction. We have added a paragraph in Introduction to highlight the adverse effects caused by salt stress on kiwifruit production from Line85-92. We agree with your suggestion to discuss the mechanistic insight on the NAC and HSF in the salt stress response. We have expanded the discussion section to include more detailed information of a possible interaction between AcNAC30 and AcHSF2b, as well as the potential role of NAC and HSF transcription factors in mediating salt stress responses in kiwifruit from Line410-421.
Best regards,
Jun Yang

Round 2
Reviewer 2 Report
Comments and Suggestions for Authors
In this revised manuscript, the authors well-addressed my previous comments.